# A Review of Reliability in Gate-All-Around Nanosheet Devices

**DOI:** 10.3390/mi15020269

**Published:** 2024-02-13

**Authors:** Miaomiao Wang

**Affiliations:** IBM Research Albany, 257 Fuller Road, Albany, NY 12203, USA; mwang@us.ibm.com

**Keywords:** gate-all-around, nanosheet, reliability, BTI, HCI, TDDB, self-heating effect, MOL, inner spacer

## Abstract

The gate-all-around (GAA) nanosheet (NS) field-effect-transistor (FET) is poised to replace FinFET in the 3 nm CMOS technology node and beyond, marking the second seminal shift in device architecture across the extensive 60-plus-year history of MOSFET. The introduction of a new device structure, coupled with aggressive pitch scaling, can give rise to reliability challenges. In this article, we present a review of the key reliability mechanisms in GAA NS FET, including bias temperature instability (BTI), hot carrier injection (HCI), gate oxide (Gox) time-dependent dielectric breakdown (TDDB), and middle-of-line (MOL) TDDB. We aim to not only underscore the unique reliability attributes inherent to NS architecture but also provide a holistic view of the status and prospects of NS reliability, taking into account the challenges posed by future scaling.

## 1. Introduction

Vertically stacked GAA NS FET, also known as multi-bridge-channel FET [1,2,3,4] and GAA nano-ribbon FET [5], represents a significant leap forward from traditional planar and FinFET devices as it offers superior electrostatics, alleviates short channel effects, provides higher effective device width per footprint, and allows flexibility in power and performance tuning with variable sheet width enabled by single-exposure EUV lithography [6,7,8,9].

The advent of GAA NS structure has not only inherited the existing reliability degradation mechanisms found in its planar and FinFET predecessors but has also introduced reliability vulnerabilities unique to its design [10,11,12,13]. Given that NS technology is progressing towards mass production and widespread industrial application, a thorough review of the NS reliability becomes imperative. This review article synthesizes the recent studies on NS reliability through both simulation and experimental methods, with the objective of giving the readers an in-depth comprehension of the unique reliability characteristics specific to NS structure as well as an all-encompassing overview of the key reliability mechanisms in GAA NS FET. We hope to shed light on areas where optimization and innovation are needed for reliability enhancement, paving the way for continued scaling and advancement of NS technology from a reliability standpoint.

The remainder of this paper is structured as follows: we begin in Section 2 by providing an extensive exploration of the specific design features of NS device architecture, including conduction surface orientation, Si channel geometry, GAA structure, and inner spacer positioned between gate and source/drain epitaxy. We analyze how these architectural aspects influence the reliability of NS FET. In Section 3, we conduct an in-depth review on each of the transistor reliability mechanisms in NS, encompassing BTI, HCI and self-heating effect (SHE), Gox TDDB, and MOL TDDB, and draw comparisons with the known reliability aspects of FinFET and planar device architectures. Gaps and challenges identified from current research works and suggestions for future research directions are discussed in Section 4. Finally, we summarize the key learnings and insights gained from this review in Section 5.

## 2. Structural Features of Nanosheet Architecture and Their Effects on Reliability

Figure 1 shows schematics of (a) a planar FET, (b) a FinFET, (c) a bulk GAA NS FET with three Si channels stacked vertically, and (d) a source-drain region cut of the bulk GAA NS FET, highlighting key components with potential impacts on reliability. In the following part of this section, we will explore how those specific components marked with blue text in Figure 1, namely conduction surface orientation, Si channel geometry, vertically stacked GAA structure, inner spacer isolation between gate and source/drain epitaxy, affect the reliability of nanosheet devices.

### 2.1. Conduction Surface Orientation

Carrier transport in planar devices is through (100) surface orientation. Contrastingly, in FinFET, carrier conduction primarily takes place through (110) sidewalls, complemented by the (100) Fin top. The predominant approach for nanosheet fabrication is to construct them on a (100) bulk Si wafer [6,7], featuring conduction mainly through (100) surface orientation of sheet top and bottom in addition to (110) side walls and arcuated corners [7,10].

GAA NS devices were fabricated on both (100) and (110) surface orientations [14], as illustrated in Figure 2. Initial interface trap densities (Dit) of GAA NS devices were extracted with AC conductance methods [15,16,17] and plotted in Figure 3. The median Dit of more than 10 NS devices on the (100) top surface is roughly 3.4 × 10^10^ cm^−2^ eV^−1^ in contrast to 9.3 × 10^10^ cm^−2^ eV^−1^ for NS devices with (110) top surface [14]. Figure 4 illustrates the comparison of NBTI degradation as a function of stress gate voltage (V_GS_) between GAA NS with (100) and (110) top surface orientations. Briefly, >1.5× worse NBTI degradation after 1000-second (s) of stress at −1.2 V was observed in NS dominated by (110) conduction, attributed to higher silicon-hydrogen (Si-H) bond density in (110) compared to (100) surface orientation [10,14]. A slightly higher activation energy (Ea) of NBTI, ~0.18 eV, was observed in Ref. [10] than in FinFET, and a higher Ea of 0.15 eV was reported in NS with (100) than that of 0.13 eV in (110) surface orientation (Figure 5) [14], owning to the different temperature dependence between hole trapping and interface trap components.

AC NBTI in NS with (100) and (110) surface orientations during alternating stress and recovery cycles are compared for different sensing delays of 1 ms and 10 ms in Figure 6 [14]. It is worth pointing out that 1.5× worse NBTI degradation is observed in NS with (110) top surface compared to (100) with both 1 ms and 10 ms sensing delay, right after 1000 s of stress and after 1000 s of recovery, showing not only higher interface trap generation but also more severe hole trapping components in (110) NS than (100).

Surface orientation effects on HCI reliability are shown in Figure 7 and Figure 8 for GAA NS nFETs and pFETs, respectively [18]. nFETs exhibit similar levels of hot carrier degradation (HCD) across both (100) and (110) surface orientations. However, pFETs show notably more severe HCD with (110) top surface than (100). The ratio of mean pFET HCD in NS with (110) top surface to that in NS with (100) top surface, HCD__110_:HCD__100_, is more than 4× after −1.2 V drain voltage (V_DS_) stress for 1000 s under high-Vg stress conditions. HCD__110_:HCD__100_ is more than 3× after −1.5 V V_DS_ stress for 1000 s under mid-Vg stress conditions. The stress gate voltage is equivalent or close to stress drain voltage under high-Vg stress conditions. The stress gate voltage is roughly between 0.5× and 0.7× of the stress drain voltage for Mid-Vg stress conditions.

It is important to note that HCD depends highly on current levels [19]. Ref. [20] reported higher electron mobility (~195 vs. ~105 cm^2^ V^−1^ s^−1^ in peak mobility) and lower hole mobility (~73 vs. ~174 cm^2^ V^−1^ s^−1^ in peak mobility) in (100) than (110) surface orientation for the HfO_2_ gate dielectric with an interfacial layer of less than 10 angstroms. The current in nFETs with the (100) surface tends to be higher than in its (110) counterpart under the same stress voltage, while the opposite is true for pFETs. Therefore, it can be concluded that HCI reliability for both NS nFETs and pFETs is generally inferior in (110) compared to (100), due to a higher Si-H bond density, leading to more interface trap generation during HCI stress.

### 2.2. Si Channel Geometry

#### 2.2.1. Impact of Tsi on Reliability and Corner Field Crowding Effect

The Si channel geometry effect on NS reliability was first observed experimentally and reported in [10]. The deterioration of PBTI and NBTI at thinner Tsi, especially for Tsi below 7 nm, was explained by the large curvature-induced corner field crowding effect [10,21]. Cross-Fin TEM images of GAA NS FETs with Tsi of 5 nm and 8 nm [10] are illustrated in Figure 9, roughly corresponding to the curvature ranges of 25~50% and 75~100%, respectively, as defined in [22] (Figure 10).

Three points are worth noting here: (1) The peak corner field dependence on curvature range reported in [22] is from the TCAD simulation of NS with different structure profiles but the same Tsi, highlighting the importance of NS corner and sidewall profile optimization for reliability improvement. (2) Although the vertical field enhancement at sheet corners is reduced when transitioning from 75% to 100% of curvature range, the proportion of the channel affected by field crowding expands, eventually encompassing the entire sidewall region. (3) Further reduction of Tsi after the 100% curvature range has been reached will result in a sharp increase in the vertical electric field, attributable to the shrinking of the radius in the curved region [21]. In addition to the higher electric field at corners than the flat sheet top and bottom, Si-H bonds, the defect precursors, are more vibrationally excited and thus easier to break, leading to higher interface trap generation at the curved regions [23].

For the same rationale as above, HCI reliability in NS degrades at thinner Tsi, as shown in Figure 11 and Figure 12, respectively [18]. For nFETs, ~1.7× of HCD is observed in NS with 4 nm of Tsi compared to that in NS with 6 nm of Tsi after high-Vg stress at 1.1 V V_DS_ for 1000 s and mid-Vg stress at 1.3 V V_DS_ for 1000 s. For pFETs, ~1.3× of HCD is observed in NS with 4 nm of Tsi compared to that in NS with 6 nm of Tsi after high-Vg stress at −1.2 V stress drain voltage for 1000 s. The observed inconsistency in the trend of HCI vs. Tsi at −1.3 V in pFETs can be attributed to the non-negligible contribution of electron trapping, which is more prominent compared to hole trapping and the generation of interface states at lower stress voltages. Note that the drastic oxide field increase at sheet corners will also affect TDDB reliability in NS [22].

#### 2.2.2. Impact of Wsheet on Reliability

Slightly degraded NBTI reliability at a narrower Wsheet is reported in NS devices fabricated on (100) substrate [10,14,24], which can be attributed to a higher contribution from (100) surface orientation and higher compressive strain at a wider Wsheet [24].

HCI reliability’s dependence on Wsheet in NS is influenced by two competing mechanisms that have conflicting effects. On the one side, the NS FET of wider sheets has a higher current and more intense self-heating effect (SHE) under the same stress condition, both contributing to an increase in HCD [12,18,25]. On the other side, the contribution from the flat areas of the sheets becomes more significant than that from the corners at wider Wsheet. HCDs in NS nFETs and pFETs with two different Wsheets are depicted in Figure 13 and Figure 14, respectively. Slightly higher HCD was observed at a wider Wsheet in both NS nFETs (1.6~2.1× of that in the narrower Wsheet under high-Vg conditions and 1~1.3× of that in narrower Wsheet under mid-Vg conditions) and pFETs (1.2~1.6× of that in the narrower Wsheet), suggesting a higher impact from the elevated current and enhanced self-heating effect [18].

### 2.3. Gate-All-Around Architecture

The continued scaling of FinFET technology beyond the 3 nm node encountered significant performance and scaling hurdles [7,26,27]. These challenges have motivated a shift from the tri-gate architecture to a vertically stackable GAA structure [27,28,29,30,31,32,33,34], aiming to mitigate short channel effects more efficiently while simultaneously boosting performance.

Among various GAA structures, NS has emerged as the leading choice and has been selected as the successor to FinFET, attributed to the higher performance achievable through wider sheets, the fabrication capability with minimal deviation from the established FinFET process, and the mitigation of some patterning complexities inherent in scaled technologies [1,2,3,4,5,6,7]. Figure 1c, d depict how multiple thin Si sheets are vertically stacked, on one top of another in a bulk NS device, to offer performance advantages over FinFET. As implied by the term “gate-all-around”, each of the Si channels in NS is encircled by high-k metal gate stacks, including an interfacial oxide (IL), a high-k dielectric layer, and the work function metals [6].

Despite the superior gate control and performance, the vertically stacked GAA structure results in increased thermal confinement, primarily due to the absence of a direct bulk connection to the Si channels and the poor thermal conductivity of the IL/high-k layers surrounding these Si channels [12]. Numerous studies from academic and industrial sources have observed a more pronounced SHE in GAA NS than in FinFET, as is evidenced by both simulation and experimental data [12,25,35,36,37,38,39,40]. SHE challenges are intensified in NS designs that feature wider and thicker sheets, and a higher count of vertically stacked Si channels [12,25,36,37,38,39]. While the transition from FinFET to NS technology may bring less thermal concern than the shift from planar to FinFET and affects only a limited portion of the circuits [12], precise thermal modeling remains crucial in NS technology for accurate HCI reliability evaluation [25,36,37,38,39].

The GAA structure is anticipated to result in a higher carrier trapping probability, subsequently leading to deteriorated HCI reliability. In GAA NS, carriers moving in all directions have the potential to be injected and become trapped in the gate oxide. In contrast, this occurs only in three directions in FinFET and just one direction in planar device [41].

### 2.4. Inner Spacer for Gate and Source/Drain Isolation

The inner spacer, the isolation between the gate and epitaxial source/drain, is a distinctive structural feature of NS FETs [6,11]. Inner spacer TDDB represents a critical reliability challenge unique to NS architecture. This issue primarily stems from the difficulties in controlling the inner spacer thickness and shape, coupled with the urgent requirement to reduce the inner spacer thickness and lower the dielectric constant (k) of the inner spacer material to enhance performance. Figure 15 illustrates the moon-shaped profile of the inner spacer in NS devices [11], which is likely to pose a higher risk of TDDB and an increased leakage concern compared to that of the top spacer situated between the poly control gate (PC) and diffusion contact (CA). Efforts in process development have been increasingly focused on achieving a more square-shaped inner spacer with improved uniformity [42], beneficial for both device performance and reliability.

### 2.5. Summary

Architectural elements discussed in Section 2 and their potential impact on NS device reliability is summarized in Table 1.

## 3. Transistor Reliability Mechanisms in Gate-All-Around Nanosheets

Recently, there has been a surge in publications exploring device reliability in NS [22,23,24,25,36,37,38,39,44,45,46]. The consensus across these studies is that the majority of the fundamental degradation mechanisms in NS devices, such as BTI, HCI, Gate oxide TDDB, and PC to CA TDDB, are similar to those in FinFET and planar devices, and governed by the same underlying physics and kinetics. Nevertheless, the unique attributes of NS, such as surface orientation and Si channel geometry, discussed in a previous section, exert a modifying effect on these degradation mechanisms.

### 3.1. BTI Reliability in NS Devices

#### 3.1.1. PBTI Reliability in NS nFETs

The shift from planar to FinFET technology has led to a significant improvement in PBTI reliability [41], owing to the decreased vertical electric field in the fully depleted device structure of FinFET compared to the bulk planar device. The PBTI advantage in FinFET over the planar device is retained when transitioning to NS technology, thanks to the preservation of the thin Si channels and thus the fully depleted device structure [6,7]. A minimal impact on PBTI reliability is anticipated from the variation in surface orientations between (100) and (110), as electron trapping is the predominant degradation mechanism, and no interface state generation is expected under moderate PBTI stress voltages. Consequently, as reported in Refs. [10,39], PBTI reliability in NS technology is comparable to that in FinFET, posing a low level of risk or concern. Note that the reduced vertical electric field in the fully depleted device structure will help to alleviate HCI and TDDB concerns in NS FET, as in FinFET [41].

#### 3.1.2. NBTI Reliability in NS pFETs

The move from planar devices to FinFETs saw a degradation in NBTI reliability, linked to the greater density of Si-H bonds and subsequently a higher rate of interface trap generation on the (110) sidewalls of FinFETs compared to the (100) surface in planar devices. Ref. [10] showed that NS exhibited comparable or better NBTI reliability compared to FinFET. Ref. [39] reported a notable, ~20% NBTI reliability improvement in their 3 nm MBCFETs than in the 4 nm and 8 nm FinFETs. Both observed improvements were attributed to the influence from surface orientation.

The NBTI reliability of NS pFETs with SiGe substrate was also investigated and compared with NS pFETs with Si substrate in [46], demonstrating that an improved NBTI in SiGe channel compared to Si can be achieved in NS pFETs with appropriate process optimization. Better NBTI reliability in the SiGe channel has been widely reported in planar and FinFET devices [47,48,49,50,51,52,53] and was attributed to compressive strain, and less accessible defects to holes in the SiGe channel.

### 3.2. HCI Reliability

Figure 16 illustrates a typical evaluation of ΔIdsat (%) with stress time during HCI stresses for GAA NS nFETs with a gate length of 12 nm [6,18].

Similar to planar devices and FinFETs [54,55,56,57,58,59,60,61], HCD in NS nFETs involves interface trap generation and electron trapping. NS nFET HCD was modeled by power law voltage and time dependence in [18]. Power law fits of ΔIdsat% versus stress time curves at various stress conditions give the time exponent (n) in the range of 0.2~0.55, with the median values of 0.25~0.4 from multiple devices at each stress condition [18]. Both voltage acceleration exponent (VAE) and n are expected to decrease with the increasing ratio of V_GS_ to V_DS_ voltages.

Representative time evolutions of HCD in NS pFETs under high-Vg and Mid-Vg stress conditions are shown in Figure 17. Note that Mid-Vg HCD in NS pFETs at low stress drain voltages no longer follows power law time dependence and is dominated by electron trapping for a short stress time, causing a current increase in contrast to the current decrease resultant from interface state generation and hole trapping [62,63].

Kim et al. reported comparable nFET HCD and worse pFET HCD in their 3 nm GAA MBCFET compared to 4 nm FINFET technology without self-heating correction [39]. After self-heating correction, the nFET HCD in 3 nm MBCFETs was slightly better than that in 4 nm FinFETs, thanks to the lower Id at reduced Vdd, and pFET HCI was comparable in 3 nm MBCFETs to 4 nm FinFETs [39].

### 3.3. Gate Oxide TDDB

Zhou et al. showed in Ref. [44] (Figure 18) that Gox TDDB in GAA NS follows Weibull statistics and Poisson area scaling with β in the range of 1.1~1.8, demonstrating robust Gox TDDB reliability in both NS nFETs and pFETs with different dipole sources.

Kim et al. also exhibited comparable Gox TDDB reliability in 3 nm MBCFETs as in 4 nm and 8 nm FinFETs with similar Weibull β distributions because of the similar EOT of those technology nodes [39].

### 3.4. MOL TDDB

The pressing need for contacted poly pitch (CPP) scaling underscores the urgency to scale both the inner spacer and top spacer thicknesses. PC to CA TDDB is reported in Ref. [39] to become worse in 3 nm MBCFETs than 4 nm and 8 nm FinFETs, mainly due to the reduction in thickness.

The conventional PC-CA TDDB test structure is built on top of the shallow trench insulator (STI) to deconvolute the impact from Gox breakdown. In contrast, the PC to Epi (inner spacer) TDDB test structure requires Si channels for source/drain epitaxial growth and needs to be built in an active region. Shen et al. proposed a novel integration scheme to evaluate inner TDDB [11] with the key process steps listed below [6,7,11]. Schematics after steps 4, 5, 6, and 7 are illustrated in Figure 19 [11].

A stack of SiGe and Si layers are epitaxially grown on the Si substrate.NS Fin revealed after Fin and STI formation.Dummy gate formation and inner spacer and junction formation.Dummy gate pull and sacrificial SiGe channel in between Si sheets are etched out.Si channel trimming to ensure final SiO_2_ thickness is closer to original Si thickness.Complete channel oxidation to avoid impact from gate oxide TDDB.HKMG formation.

By fully oxidizing the silicon channel to push the breakdown of Gox to a much higher voltage than that of inner spacer, the Vmax of 1.3 V and Emax of 3 MV/cm are projected for the inner spacer TDDB at 125 °C with a 2500 m run length and 100 ppm failure rate [11].

In Figure 20, using the β value, the time to 63% fail (T63%), VAE reported in [11,44], the time to failure of Gox nFETs and the inner spacer are projected to a specified failure rate and target area (for Gox) and run length (for inner spacer), and plotted as a function of stress voltage. Due to the shallower β and lower VAE, the inner spacer of NS is more prone to failure compared to gate oxide at voltage closing to standard operating conditions, especially when a stringent low failure rate target is required. Scaling the inner spacer thickness for future technology nodes poses significant challenges to TDDB reliability. Achieving uniformity in both thickness and shape and the profile optimization of the inner spacer are crucial for success in this endeavor [11,42].

### 3.5. Summary

Key modeling parameters for transistor reliability mechanisms in GAA NS reported in the recent literature are summarized in Table 2 and Table 3, below.

## 4. Reliability Challenges in NS FETs and Gaps for Future Learning

Based on the discussions earlier, MOL TDDB, especially inner spacer TDDB, presents significant reliability challenges in NS technology. Process innovation in inner spacer shape optimization, uniformity control, and material innovation for enhanced TDDB robustness at lower k are essential, particularly in the context of pushing the boundaries of continuous scaling in NS technology.

As channel lengths are reduced while the current increases, HCI is expected to worsen, posing considerable concern for future scaling.

Despite the recent surge in reliability research for NS devices, there remain areas and aspects where studies are either lacking or absent. Notably, this includes investigations into the TDDB reliability of substrate isolation and its impact (Figure 21) on NS reliability and thermal property [6,7,9], the effect of Tsus, which refers to the spacing between Si channels, the impact of multi-Vt and dipole processes on BTI and HCI reliability, the reliability impact from quantum confinement [6,9,64,65], reliability variability and concerns arising from the non-uniformity of thermal and electrical properties across different sheets [66], and inner spacer and top spacer reliability with different spacer materials and MOL integration schemes. These under-explored areas are critical for a more comprehensive understanding of NS reliability.

## 5. Conclusions

In this article, we conduct an exhaustive review of the device reliability mechanisms in vertically stacked GAA NS FETs. We reveal that, apart from the novel failure mode of inner spacer TDDB, conventional reliability degradation mechanisms, such as BTI, HCI, gate oxide TDDB, and PC to CA TDDB in NS devices are akin to those in FinFET and planar architectures. We highlight the significant influence of Si channel geometry and the profile of corners and sidewalls on NS reliability, underlining the importance of considering reliability factors in the design of the NS process and structure. We pinpoint inner spacer TDDB, PC to CA TDDB, and HCI as major hurdles for the continued scaling and advancement of NS technology. Furthermore, we suggest areas for future exploration to encompass the full spectrum of reliability vulnerabilities in NS technology.

## Figures and Tables

**Figure 1 micromachines-15-00269-f001:**
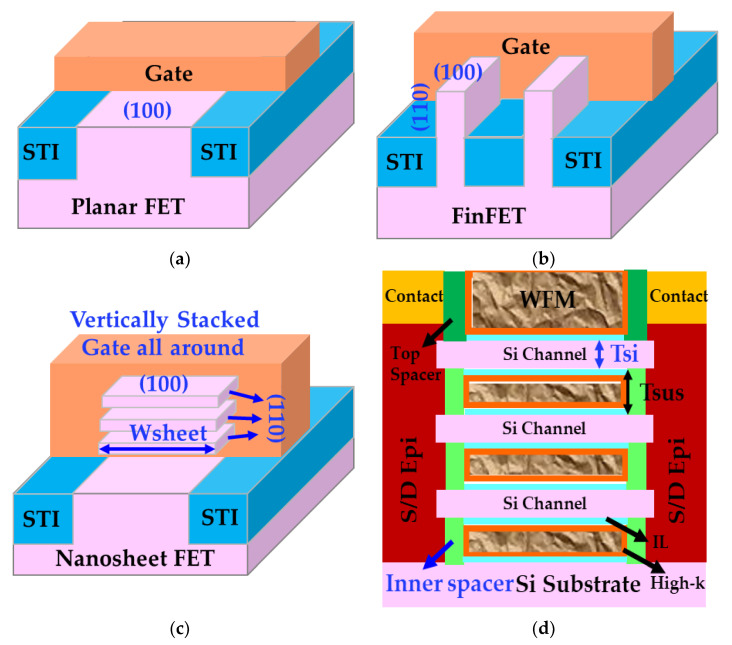
Schematics of (**a**) a planar device, (**b**) a FinFET, (**c**) a vertically stacked bulk GAA NS FET, and (**d**) a cut of bulk GAA NS FET across the source-drain region where the key components marked with blue text are: surface orientations of Si channels in a planar FET, a FinFET, and a bulk NS FET, respectively, the thickness of the NS Si channels (Tsi), the width of the NS Si channels (Wsheet), GAA architecture, and inner spacers for gate and source/drain isolation physically.

**Figure 2 micromachines-15-00269-f002:**
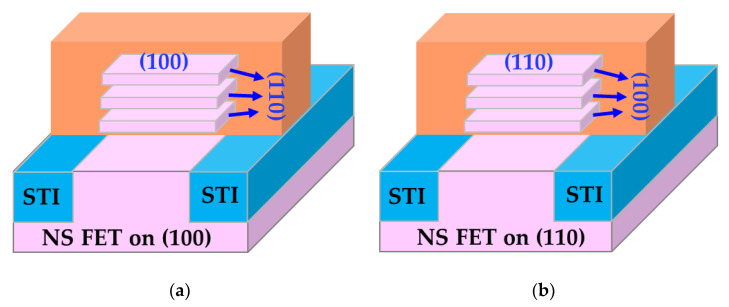
Schematics of GAA NS FETs fabricated on (**a**) (100) and (**b**) (110) surface orientations.

**Figure 3 micromachines-15-00269-f003:**
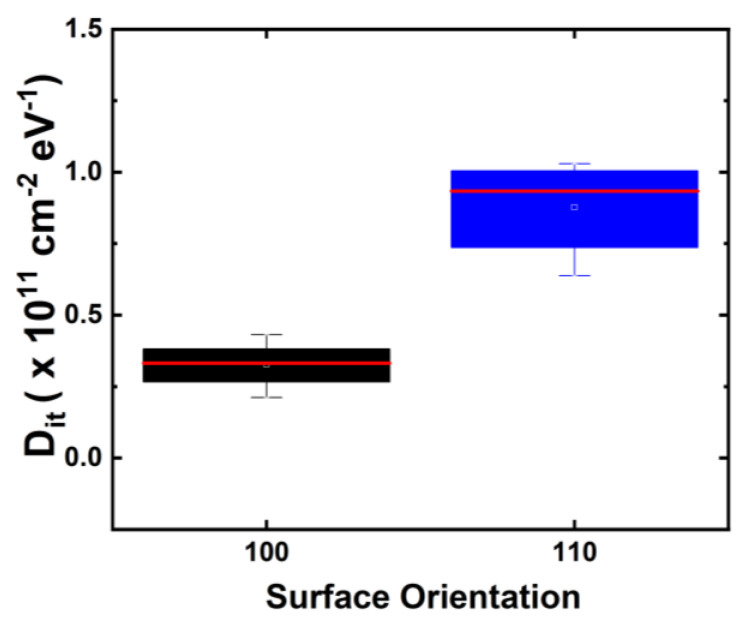
Comparison of Dit levels in GAA NS fabricated on (100) vs. (110) surface orientation extracted with AC conductance method [15,16,17], showing a higher initial Dit in NS FETs with (110) top surface than (100) [14], both lower than 1 × 10^11^ cm^−2^ eV^−1^. Reprinted/adapted with permission from IEEE Proceedings of the 2020 International Reliability Physics Symposium.

**Figure 4 micromachines-15-00269-f004:**
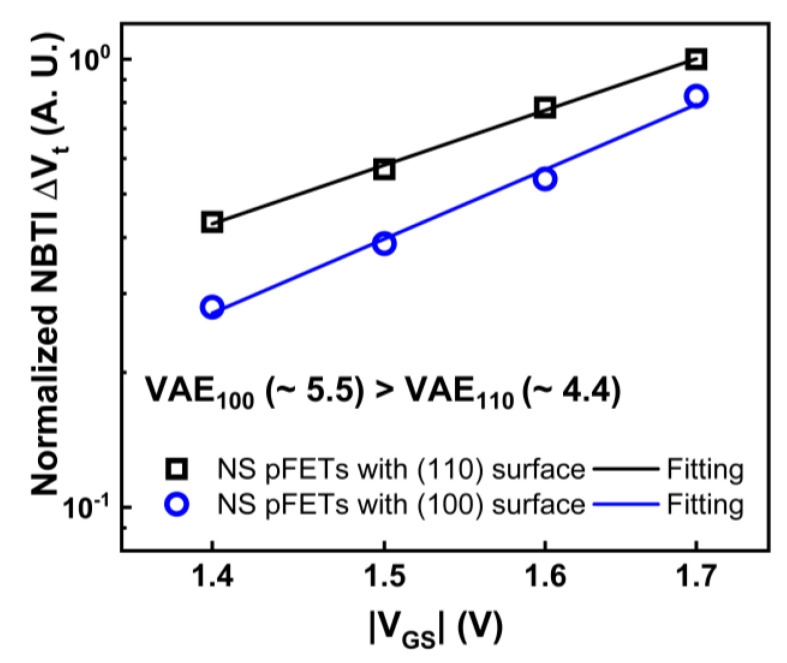
Comparison of NBTI-induced Vt shift (ΔVt) as a function of absolute stress gate voltage (|V_GS_|) in GAA NS devices fabricated on (100) vs. (110) surface orientations, showing higher NBTI degradation in (110) surface orientation [14]. Reprinted/adapted with permission from IEEE Proceedings of the 2020 International Reliability Physics Symposium.

**Figure 5 micromachines-15-00269-f005:**
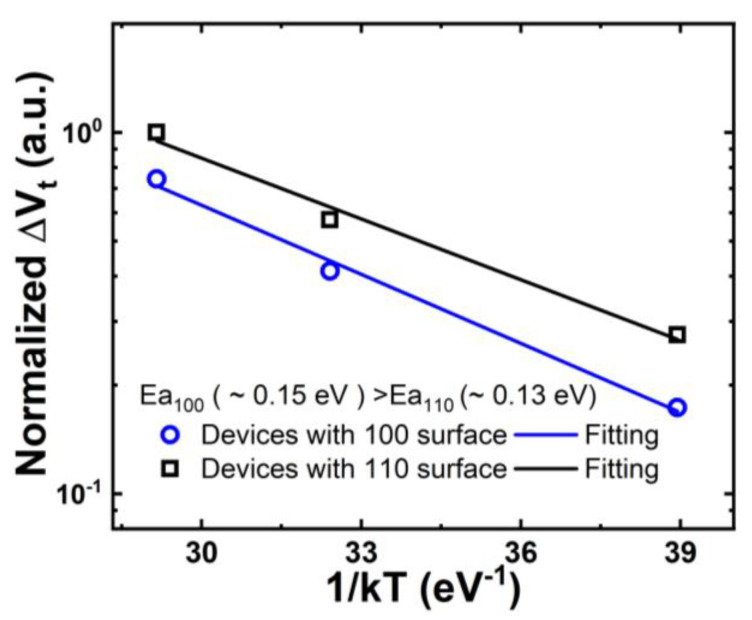
Comparison of activation energy, Ea, of NBTI in GAA NS fabricated on (100) vs. (110) surface orientations, showing a higher Ea in (100) than (110) surface orientation [14]. Reprinted/adapted with permission from IEEE Proceedings of the 2020 International Reliability Physics Symposium.

**Figure 6 micromachines-15-00269-f006:**
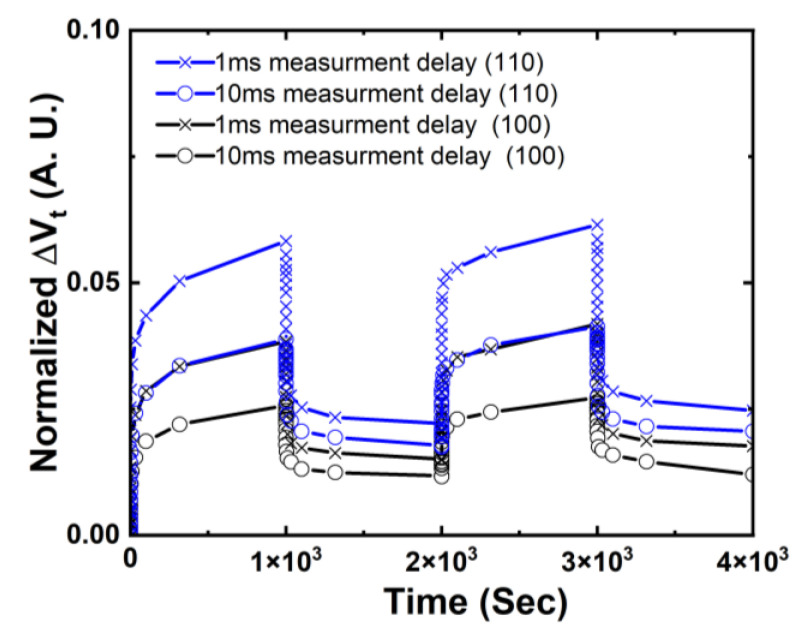
Comparison of NBTI-induced ΔVt for GAA NS FETs fabricated on (100) vs. (110) surface orientations during alternating stress and recovery cycles of 1000 s each. The total stress and recovery time is 4000 s for each device [14]. Impact from sensing delay of 1 ms vs. 10 ms was also shown and discussed [14]. It is worth highlighting that, in addition to a higher generation of interface traps, more hole trapping was observed in (110) surface orientation. This was evident from the increased magnitude of ΔVt recovery difference between 1 ms and 10 ms for (110) surface orientation, implying that the recoverable defect trapping captured by 1 ms sense delay but discharged during 10 ms sense delay was higher at (110) than (100) surface orientation. Reprinted/adapted with permission from IEEE Proceedings of the 2020 International Reliability Physics Symposium.

**Figure 7 micromachines-15-00269-f007:**
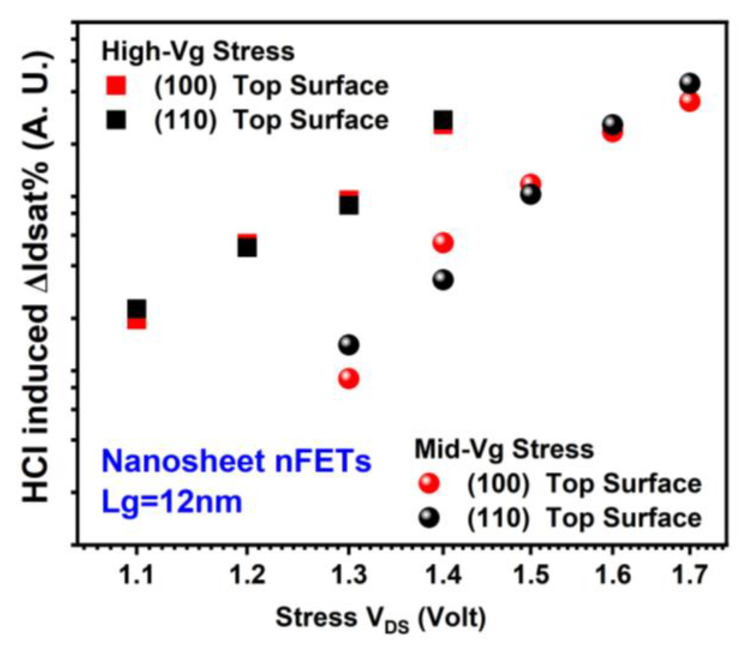
Comparison of HCD (each data point is the mean value of more than seven devices stressed at the same condition) in GAA NS nFETs with (100) vs. (110) top surface orientations [18]. ΔIdsat% is defined as (Idsat0 – Idsat)/Idsat0 × 100%, whereas Idsat0 is the initial saturation drain current. Idsat is the saturation drain current during stress.

**Figure 8 micromachines-15-00269-f008:**
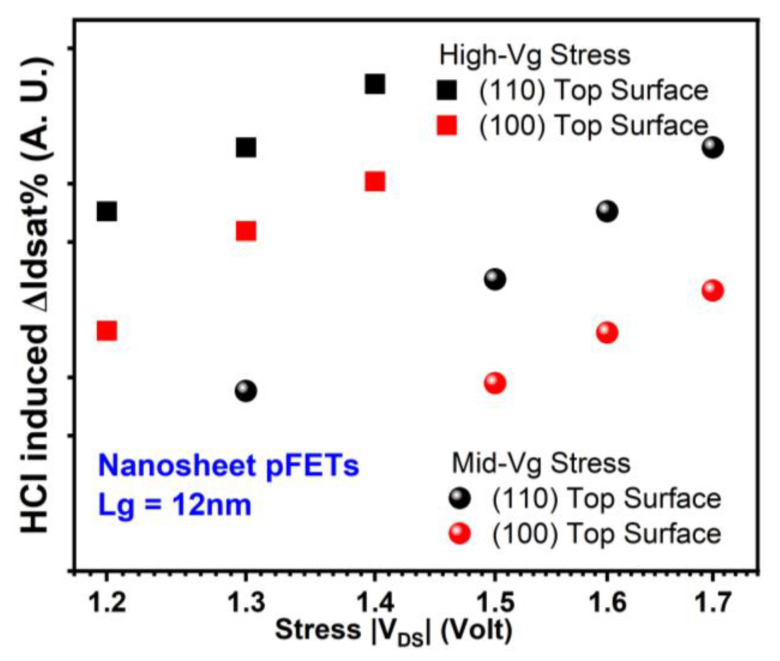
Comparison of HCD (each data point is the mean value of more than seven devices stressed at the same condition) in GAA NS pFETs with (100) vs. (110) top surface orientations [18].

**Figure 9 micromachines-15-00269-f009:**
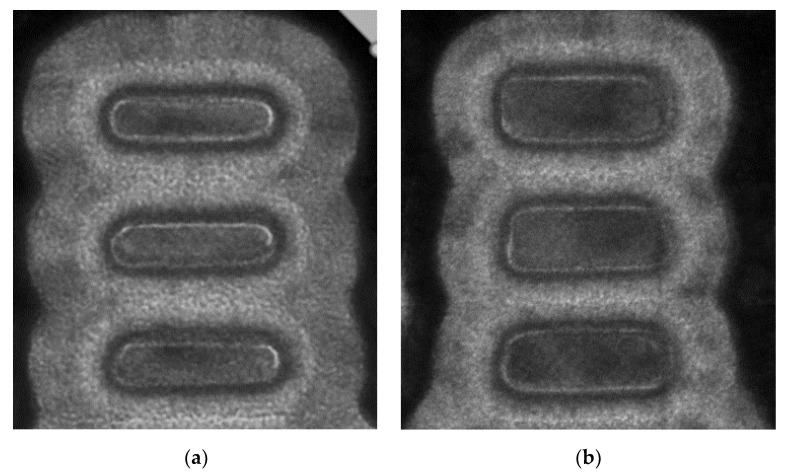
Cross-Fin TEMs of vertically stacked GAA NS devices with Tsi of approximately (**a**) 5 nm and (**b**) 8 nm, corresponding to the curvature ranges of 25~50% and 75~100%, respectively, as defined in Figure 10 [10]. Reprinted/adapted with permission from IEEE Proceedings of the 2019 International Reliability Physics Symposium.

**Figure 10 micromachines-15-00269-f010:**
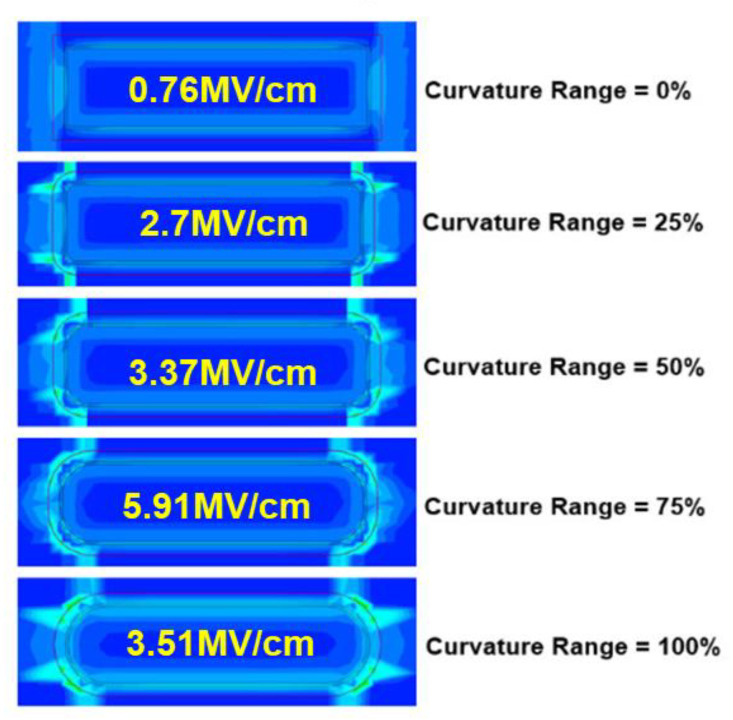
Vertical electric field peaks at sheet corners for GAA NS with different structure (sidewalls and corners) profiles and curvature ranges from TCAD simulation [22]. The vertical field at sheet corners increases as the curvature range changes from 0% to 75% and then reduces when transitioning from 75% to 100% of curvature range. Reprinted/Adapted from [22], under a Creative Commons Attribution-NonCommercial-NoDerivatives 4.0 License. Source: https://creativecommons.org/licenses/by-nc-nd/4.0/. Modifications were made to the original figure.

**Figure 11 micromachines-15-00269-f011:**
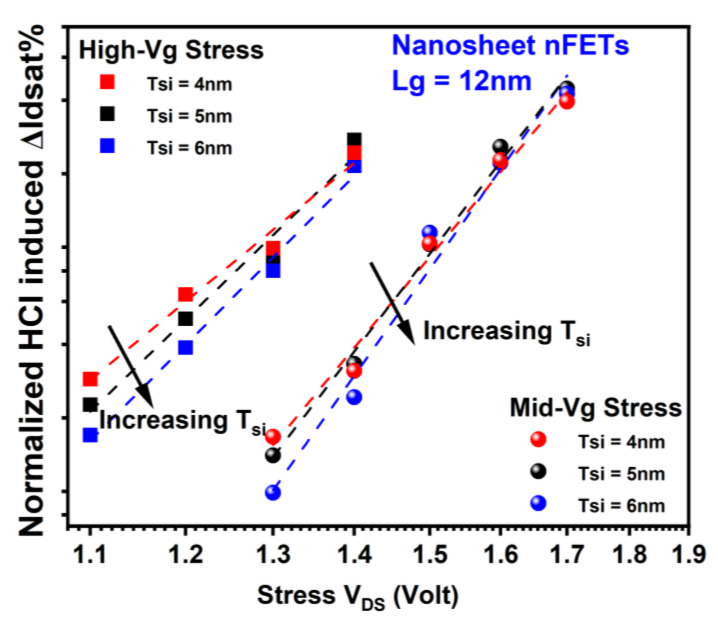
HCD as a function of stress drain voltage in GAA nFETs for different Tsi. Each data point is the mean value of more than seven devices stressed at the same condition. Enhanced HCI damage at thinner Tsi can be attributed to a higher corner field at scaled diameters of the curved region [18,21,22]. Stress gate voltage is equivalent or close to stress drain voltage under high-Vg stress conditions. Stress gate voltage is roughly between 0.5× and 0.7× of stress drain voltage for Mid-Vg stress conditions.

**Figure 12 micromachines-15-00269-f012:**
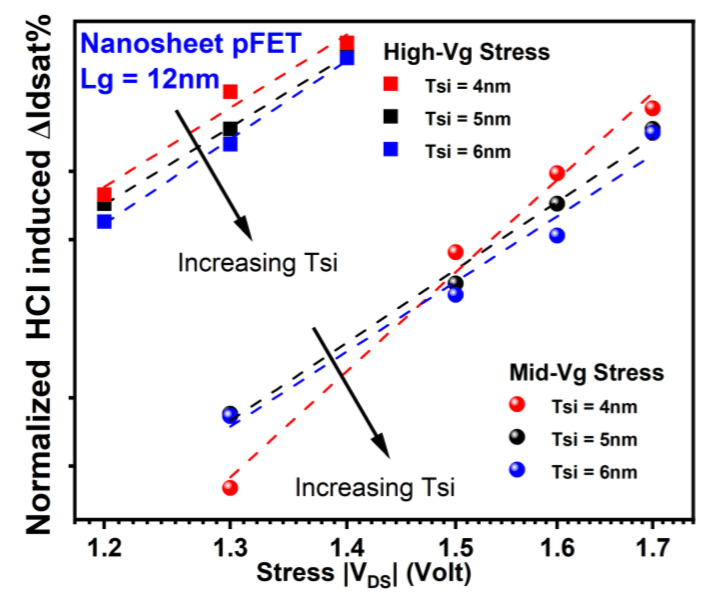
HCD as a function of stress drain voltage in GAA pFETs for different Tsi. Each data point is the mean value of more than seven devices stressed at the same condition. Enhanced HCI damage at thinner Tsi can be attributed to more severe corner field crowding effect at scaled diameters of the curved region [18,21,22]. Stress gate voltage is equivalent or close to stress drain voltage under high-Vg stress conditions. Stress gate voltage is roughly between 0.5× and 0.7× of stress drain voltage for Mid-Vg stress conditions.

**Figure 13 micromachines-15-00269-f013:**
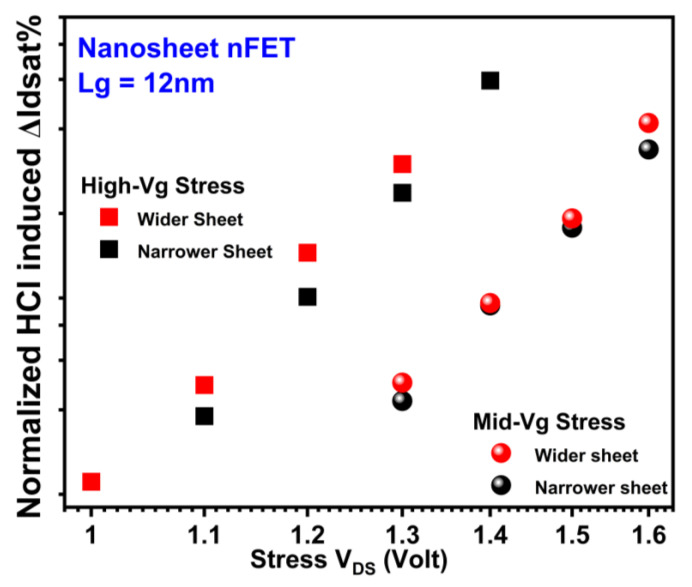
Wsheet dependence of HCI reliability in GAA NS nFETs with wider sheets exhibits higher HCD due to enhanced current and SHE [18]. Each data point is the mean value of multiple devices stressed at the same condition.

**Figure 14 micromachines-15-00269-f014:**
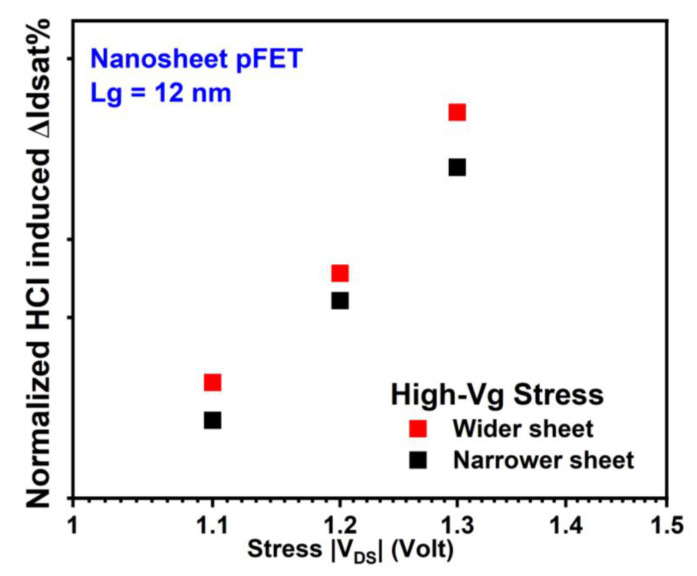
Wsheet dependence of HCI reliability in GAA NS pFETs with wider sheets exhibits higher HCD due to enhanced current and SHE [18]. Each data point is the mean value of multiple devices stressed at the same condition.

**Figure 15 micromachines-15-00269-f015:**
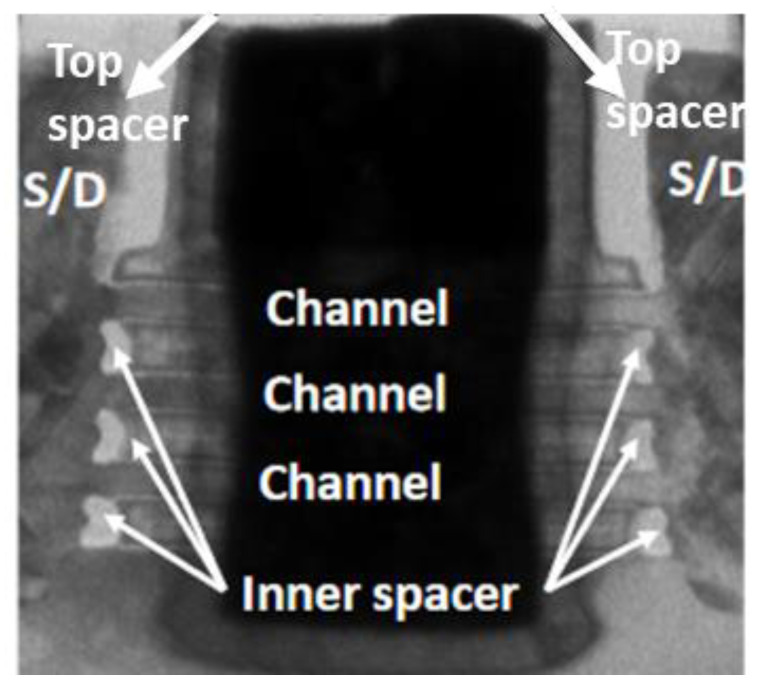
Cross-sectional TEM image of an NS device highlighting top spacers and moon-shaped inner spacers between gate and source/drain epitaxy [11]. Reprinted/adapted with permission from IEEE Proceedings of the 2020 International Reliability Physics Symposium.

**Figure 16 micromachines-15-00269-f016:**
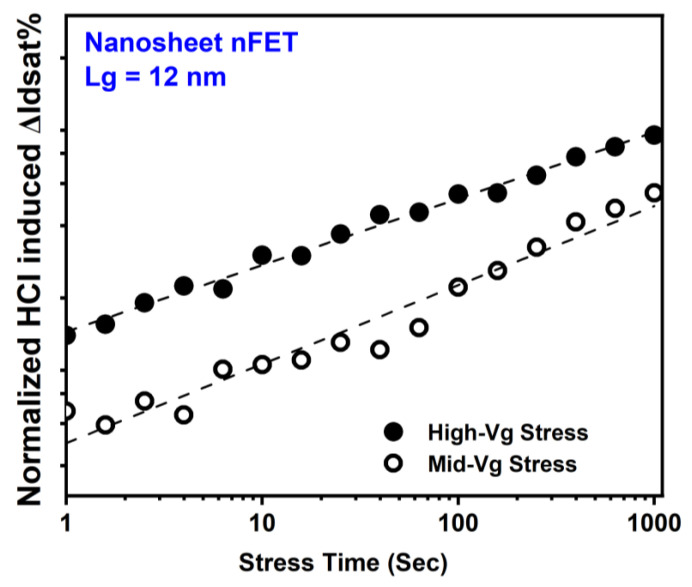
Typical HCD as a function of stress time in GAA NS nFETs with Lg = 12 nm under high-Vg and Mig-Vg HC stresses, following power law time dependence [18].

**Figure 17 micromachines-15-00269-f017:**
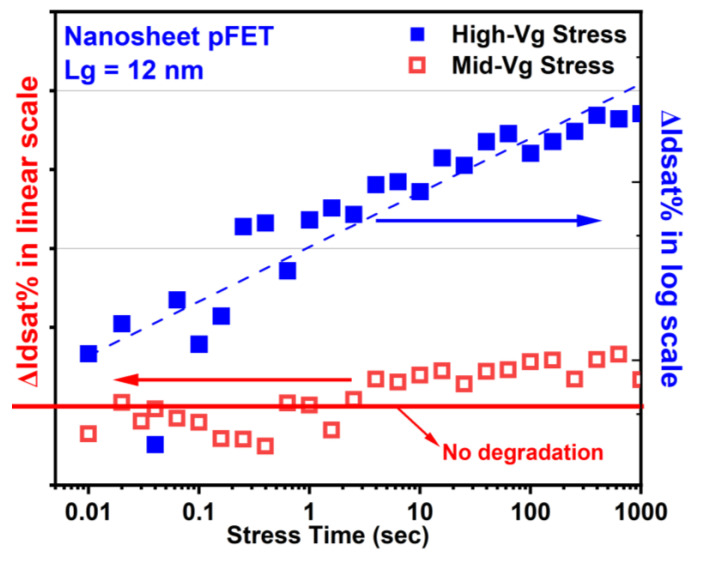
Typical HCD as a function of stress time in GAA NS pFETs with Lg = 12 nm under high-Vg and Mig-Vg HC stress conditions. High-Vg HCD in NS pFETs follows power law time dependence [18]. Mid-Vg HCD in NS pFETs at low stress drain voltages no longer follows power law time dependence and is dominated by electron trapping for a short stress time, causing a current increase in contrast to the current decrease resultant from interface state generation hole trapping [62,63]. Stress gate voltage is equivalent or close to stress drain voltage under high-Vg stress conditions. Stress gate voltage is roughly between 0.5× and 0.7× of stress drain voltage under Mid-Vg stress conditions.

**Figure 18 micromachines-15-00269-f018:**
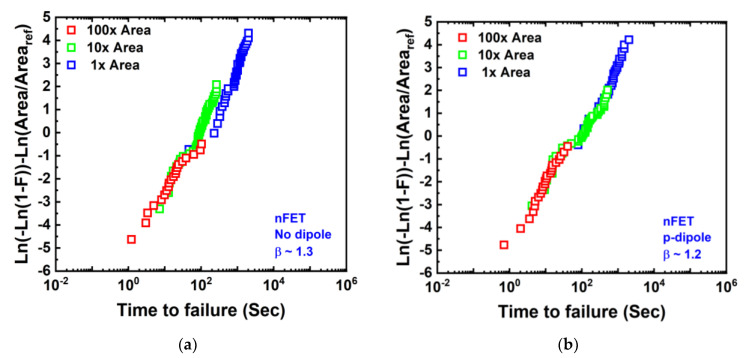
Area scaling comparison for NS nFETs with (**a**) no-dipole, (**b**) p-dipole source, and (**c**) n-dipole source in gate stacks and NS pFETs with (**d**) no-dipole, (**e**) p-dipole source, and (**f**) n-dipole source in gate stacks. All follow Weibull statistics and Poisson area scaling statistics [44]. Reprinted/adapted with permission from IEEE Proceedings of the 2021 International Reliability Physics Symposium.

**Figure 19 micromachines-15-00269-f019:**
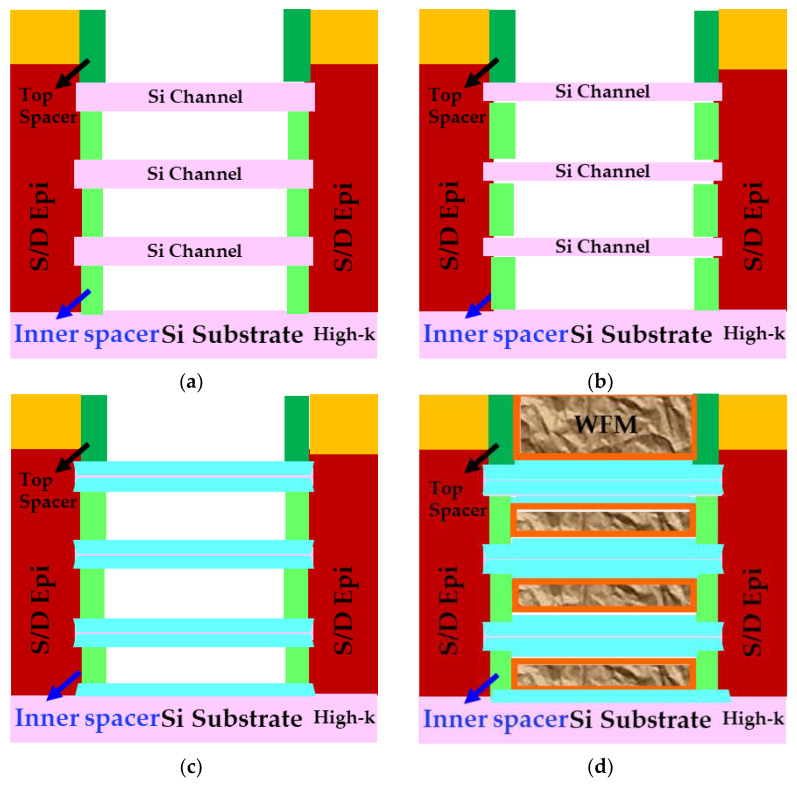
Cross-sectional schematics after key steps in the special process flow in [11] to evaluate inner spacer TDDB: (**a**) after channel release, (**b**) after Si trimming, (**c**) complete channel oxidation, and (**d**) after HKMG process. Reprinted/adapted with permission from IEEE Proceedings of the 2020 International Reliability Physics Symposium.

**Figure 20 micromachines-15-00269-f020:**
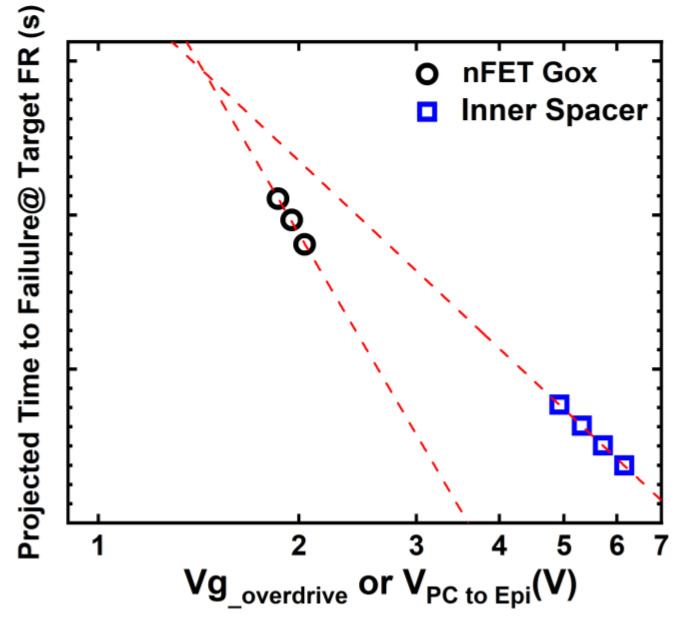
Time to failures of Gox TDDB in NS nFETs and inner spacer TDDB projection to a target failure rate, and total Gox area or inner spacer run length, with the T63%, VAE, and β reported in [11,44]. Due to the shallower beta and lower VAE, the inner spacer of NS is likely to fail sooner than gate oxide at maximum operating voltage and the required failure rate for standard semiconductor chip operation. Adapted with permission from IEEE Proceedings of the 2020 and 2021 International Reliability Physics Symposium.

**Figure 21 micromachines-15-00269-f021:**
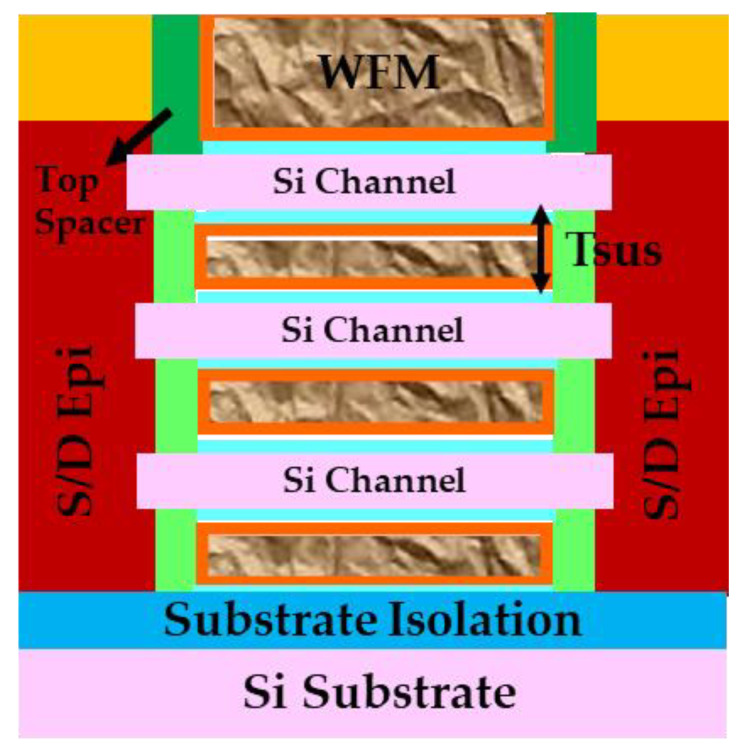
A cut of GAA NS FET across the source-drain region with substrate isolation [9]. The spacing between Si channels, Tsus, with potential impact on HKMG metal fill, and thereby gate stack reliability, is also highlighted in the schematic.

**Table 1 micromachines-15-00269-t001:** Summary of key architectural elements of NS and their impact on device reliability. BT. stands for better than; * NIE stands for no impact expected; and ** NR is short for not reported. *** Expected from the corner field and surface orientation impact at different sheet widths.

Mechanisms	PBTI	NBTI	HCI	SHE	Gox TDDB	MOL TDDB
Surface orientation	NIE *	Yes	Yes	NIE *	Yes	NIE *
(100) BT. (110)?	[10]	[10,14]	[18]		[43]	
Tsi: 9 nm and below	Yes	Yes	Yes	Worse	Yes	NR **
Thicker BT. Thinner?	[10]	[10]	[18,23]	[38]	[22]	
Wsheet	NIE *	Yes	Worse	Worse	Yes ***	NR **
Wider BT. Narrow?		[10,14]	[12,18,24]	[12,25,36]	[10,43]	

**Table 2 micromachines-15-00269-t002:** Key modeling parameters for BTI and HCI reliability in GAA NS [10,14,18,36]. * Extracted from power law fitting of HCD vs. stress voltage curves in NS with (100) top surface orientation in Figure 4.

Mechanisms	References	Temp	VAE from Power Law Fit	Time Exponent (n)	Activation Energy (Ea)
PBTI	[10]	125 °C	~7.4	~0.20	0.105 eV
	[36]	25~125 °C	8.61~10.18	-	-
NBTI	[10]	125 °C	~5.5	~0.25	0.18 eV
	[14]		~5.52 for (100)		0.15 eV for (100)
			~4.40 for (110)		0.13 eV for (110)
nFET Mid-Vg HCI	[18]	25 °C	~13.2 *	0.25~0.4	0.07 eV
nFET High-Vg HCI	[18]	25 °C	~10.3 *		0.07 eV
pFET Mid-Vg HCI	[18]	25 °C	~8.8 *	-	-
pFET High-Vg HCI	[18]	25 °C	~11.6 *	-	0.17 eV

**Table 3 micromachines-15-00269-t003:** Key modeling parameters for Gox and inner spacer reliability in GAA NS [11,44]. Adapted with permission from IEEE Proceedings of the 2020 and 2021 International Reliability Physics Symposium.

Mechanisms	Dipole Process	References	Temp.	VAE	β	Activation Energy (Ea)
nFET Gox TDDB	no Dipole	[44]	125 °C	57	1.3	0.81 eV
nFET Gox TDDB	p-Dipole	[44]	125 °C	57	1.2	0.70 eV
nFET Gox TDDB	n-Dipole	[44]	125 °C	62	1.8	0.64 eV
pFET Gox TDDB	no Dipole	[44]	125 °C	44	1.3	0.55 eV
pFET Gox TDDB	p-Dipole	[44]	125 °C	51	1.1	0.59 eV
pFET Gox TDDB	n-Dipole	[44]	125 °C	45	1.1	0.83 eV
Inner spacer TDDB		[11]	25 °C (RVS)	52	0.6	0.54 eV
			125 °C (RVS)	31	0.8	
			125 °C (CVS)	32.4	0.57	

## Data Availability

Not applicable.

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
