# Peer review of "A Review of Reliability in Gate-All-Around Nanosheet Devices"

_micromachines, 2024, doi:10.3390/mi15020269_

Round 1

Reviewer 1 Report

Comments and Suggestions for Authors

This review article discusses the impact of Gate-all-around nano sheet FET structure as a highly viable device architecture to emerge in the field in the point of view of reliability in various mechanisms and parameters that are key to the device structure. The review article is well-written and the structure of the discussion is logical and well-organized. The only thing the author needs to address is in Table 1, the comparisons of the impact between various NS architectural elements should be quantified or at the very least more descriptive.  At its current form "better" or "worse" is confusing to the general reader especially in rows 2 and 3: is thicker "better" than thinner, or is wider "better" than narrow? Other that this issue, I think the manuscript only needs minor revisions and spelling check and it should be considered for publication in Micromachines.

Comments on the Quality of English Language

The entire manuscript needs to be canned for spelling (example: the abstract first sentence "Gall-all-around" should be "GATE-all-around"). There are a number of these mistakes in the manuscript that need to be corrected.

Author Response

Dear Reviewer:

I would like to express my sincere appreciation for your review of our manuscript. Your valuable feedback has been instrumental in improving the quality of our work.  We have made the recommended revisions and improvements based on your suggestions and we believe these changes have strengthened our manuscript.

Review 1’s  Comments and Suggestions for Authors:

This review article discusses the impact of Gate-all-around nano sheet FET structure as a highly viable device architecture to emerge in the field in the point of view of reliability in various mechanisms and parameters that are key to the device structure. The review article is well-written, and the structure of the discussion is logical and well-organized.

The only thing the author needs to address is in Table 1, the comparisons of the impact between various NS architectural elements should be quantified or at the very least more descriptive.  At its current form "better" or "worse" is confusing to the general reader especially.  In rows 2 and 3: is thicker "better" than thinner, or is wider "better" than narrow? Other that this issue, I think the manuscript only needs minor revisions and spelling check and it should be considered for publication in Micromachines.

Response: Thank you very much for the helpful feedback. We have revised the format of Table 1 to enhance the clarity.  We have also added quantified comparisons between HCD and BTI degradation between  (100) vs (110) surface orientations, thicker vs thinner Si channel thicknesses, and Wider vs narrower Wsheets in section 2.1 and 2.2 (red text highlighted in yellow).

Review 1’s  Comments on the Quality of English Language:

The entire manuscript needs to be canned for spelling (example: the abstract first sentence "Gall-all-around" should be "GATE-all-around"). There are a number of these mistakes in the manuscript that need to be corrected.

Response:  Thank you very much for the thorough review of our manuscript. We have carefully checked the spellings and corrected the typos ( red text highlighted in yellow).

Once again thank you very much for your time and insightful comments.

Sincerely.

Miaomiao Wang

Reviewer 2 Report

Comments and Suggestions for Authors

The manuscript presents a very timely review of reliability in gate-all-around nanosheet devices. It is well written and presented in a very clear fashion. This reviewer recommends publishing the manuscript in present form.

Author Response

Dear Reviewer:

Thank you very much for your positive feedback and affirmation regarding our manuscript.  We are delighted that our work resonated with you , and we value the time you invested in reviewing our manuscript. Your encouragement inspires us to continue striving for excellence.

Best regards,

Miaomiao Wang

Reviewer 3 Report

Comments and Suggestions for Authors

The review provides a solid foundation for discussing the reliability mechanisms of GAA NS FET but could benefit from improvements in clarity, conciseness, and engaging the reader more effectively. The review is timely and could be accepted for publication following the minor revision:

Abstract, first word, Gall-all-around (GAA), should be Gate-all-around.

The text provides a comprehensive discussion with qualitative analysis, a more quantitative discussion related to the literature should be discussed.

Further, the sub-sections and sections from introduction onwards are not connected with each other.

Line 123, quantitatively how much is the electron and hole mobility. Can it be influenced by temperature and pressure? Mention the measurement conditions?

There are many sentences, where the authors mention quantitative value and refer to a reference. Kindly provide the exact values. Such as line 76-79, 172

Conclusion should provided with a perspective based on the discussed results to lead the researchers.

Author Response

Dear Reviewer:

Thank you sincerely for your review and insightful comments on our manuscript. Your constructive input is invaluable in refining our manuscript. We carefully addressed each of your points to enhance the quality of our work.

The review provides a solid foundation for discussing the reliability mechanisms of GAA NS FET but could benefit from improvements in clarity, conciseness, and engaging the reader more effectively. The review is timely and could be accepted for publication following the minor revision:

Abstract, first word, Gall-all-around (GAA), should be Gate-all-around.

Response: Thank you very much for the thorough review of our manuscript. We have corrected the typos ( red text highlighted in yellow).

The text provides a comprehensive discussion with qualitative analysis, a more quantitative discussion related to the literature should be discussed.

Line 123, quantitatively how much is the electron and hole mobility. Can it be influenced by temperature and pressure? Mention the measurement conditions?

Response:  Yes, mobility is highly temperature dependent due to coulomb scattering.  We have added the following descriptions and Reference [20] from L. Chang et al. in the manuscript for a quantitative comparison of the electron and hole mobilities in (100) vs (1a0) surface orientations in HfO2 gate dielectric with thin (<10A) interfacial layer.  Ambient conditions ( ~ Room T and ~ 1 atm ) should be used in Red [20] as no pressure and temperature conditions were specified.

“Ref.[20] reported higher electron mobility ( ~ 195 vs ~ 105 cm2*V-1s-1 in peak mobility) and lower hole mobility (~ 73 vs ~ 174 cm2*V-1s-1 in peak mobility) in (100) than (110) surface orientation for HfO2 gate dielectric with an interfacial layer of less than 10 angstroms.

There are many sentences, where the authors mention quantitative value and refer to a reference. Kindly provide the exact values. Such as line 76-79, 172.

Response: Thanks very much for your helpful feedback. We acknowledge the importance of providing exact (instead of normalized values) to enhance the technical strength of the paper. We have added as many exact values as available in the revised manuscript ( red text highlighted in yellow) in section 2.  However,  obtaining additional information if not already included in the manuscript sometimes be very challenging. Even for the papers from our team, disclosing absolute values is often restricted by confidentiality considerations, preventing internal approval for publication. In lieu of exact values, we have also added in the manuscript as many quantitative comparisons as possible in section 2  ( red text highlighted in yellow).

Conclusion should provided with a perspective based on the discussed results to lead the researchers. 

Response:  Thanks very much for your helpful input. The conclusions that will help to lead the researchers are summarized below:

  • Si channel geometry and shape have strong impact on reliability and need to be considered for design and reliability co-optimization: “We highlight the significant influence of Si channel geometry and profile of corners and sidewalls on NS reliability, underlining the importance of considering reliability factors in the design of NS process and structure” in section 5.
  • We provide insight on the most challenging reliability mechanisms for NS scaling: “ We pinpoint inner spacer TDDB, PC to CA TDDB, and HCI as major hurdles for continued scaling and advancement of NS technology.”
  • We suggest areas for future exploration to encompass the full spectrum of reliability vulnerabilities in NS technology in section 4. “ this includes investigations into TDDB reliability of substrate isolation and its impact ( Figure 21) on NS reliability and thermal property [7, 9], the effect of Tsus, which refers to the spacing between Si channels, impact of multi-Vt and dipole process on BTI and HCI reliability, reliability impact from quantum confinement [7, 9, 64, 65], reliability variability and concerns arising from the non-uniformity of thermal and electrical properties across different sheets [66], and inner spacer and top spacer reliability with different spacer materials and MOL integration schemes. These under-explored areas are critical for a more comprehensive understanding of NS reliability.”

Further, the sub-sections and sections from introduction onwards are not connected with each other.

Response:  Thanks very much for your helpful feedback.  The objective of this work is to review all transistor reliability aspects of NS device architecture to see whether there is any fundamental concern of qualifying nanosheet as a viable technology and its scalability from a reliability standpoint.  The sections and sub-sections of this paper after introduction is organized as follows: In section 2, we review all the specific features of NS structure and their potential impact on reliability including 2.1 surface orientation, 2.2  Si channel geometry, 2.3 GAA structure, and 2.4 inner spacer respectively.  In section3, we review all the transistor reliability mechanisms inherent to NS devices, including 3.1 BTI, 3.2 HCI, 3.3 GoxTDDB and 3.4 MOL TDDB ( PC to CA TDDB and PC to Epi TDDB) respectively.   Section 4 discusses the reliability challenges NS device is facing and gaps requires future study and improvement. Section 5 summarizes the conclusions from the previous sections:

Once again thank you very much for your valuable and helpful comments.

Sincerely.

Miaomiao Wang
